# Structural and biochemical characterization of the key components of an auxin degradation operon from the rhizosphere bacterium *Variovorax*

**Yongjian Ma**[1☯], **Xuzichao Li**[1☯], **Feng Wang**[2☯], **Lingling Zhang**[1], **Shengmin Zhou**[3], **Xing Che**[3], **Dehao Yu**[4], **Xiang Liu**[5], **Zhuang Li**[2]*, **Huabing Sun**[4]*, **Guimei Yu**[1]*, **Heng Zhang**[1]*

**1** State Key Laboratory of Experimental Hematology, Key Laboratory of Immune Microenvironment and Disease (Ministry of Education), The Province and Ministry Co-sponsored Collaborative Innovation Center for Medical Epigenetics, Haihe Laboratory of Cell Ecosystem, Department of Biochemistry and Molecular Biology, School of Basic Medical Sciences, Tianjin Medical University, Tianjin, China, **2** State Key Laboratory of Biocatalysis and Enzyme Engineering, School of Life Sciences, Hubei University, Wuhan, China, **3** YDS Pharmatech, Albany, New York, United States of America, **4** The Province and Ministry Co-sponsored Collaborative Innovation Center for Medical Epigenetics, Tianjin Medical University; Tianjin Key Laboratory on Technologies Enabling Development of Clinical Therapeutics and Diagnostics, School of Pharmacy, Tianjin Medical University, Tianjin, China, **5** State Key Laboratory of Medicinal Chemical Biology, Frontiers Science Center for Cell Responses, College of Life Sciences, Nankai University, Tianjin, China

☯ These authors contributed equally to this work.
* zhuangli@hubu.edu.cn (ZL); sunhuabing@tmu.edu.cn (HS); gyu324138@gmail.com (GY); zhangheng134@gmail.com (HZ)

**Data Availability Statement:** The atomic coordinates of the structures have been deposited in the Protein Data Bank under accession codes 7YLT (IadK2), 7YLS (IadD/E), 7YLR (IadC) and

## Abstract

Plant-associated bacteria play important regulatory roles in modulating plant hormone auxin levels, affecting the growth and yields of crops. A conserved auxin degradation (*iad*) operon was recently identified in the *Variovorax* genomes, which is responsible for root growth inhibition (RGI) reversion, promoting rhizosphere colonization and root growth. However, the molecular mechanism underlying auxin degradation by *Variovorax* remains unclear. Here, we systematically screened *Variovorax iad* operon products and identified 2 proteins, IadK2 and IadD, that directly associate with auxin indole-3-acetic acid (IAA). Further biochemical and structural studies revealed that IadK2 is a highly IAA-specific ATP-binding cassette (ABC) transporter solute-binding protein (SBP), likely involved in IAA uptake. IadD interacts with IadE to form a functional Rieske non-heme dioxygenase, which works in concert with a FMN-type reductase encoded by gene *iadC* to transform IAA into the biologically inactive 2-oxindole-3-acetic acid (oxIAA), representing a new bacterial pathway for IAA inactivation/degradation. Importantly, incorporation of a minimum set of *iadC/D/E* genes could enable IAA transformation by *Escherichia coli*, suggesting a promising strategy for repurposing the *iad* operon for IAA regulation. Together, our study identifies the key components and underlying mechanisms involved in IAA transformation by *Variovorax* and brings new insights into the bacterial turnover of plant hormones, which would provide the basis for potential applications in rhizosphere optimization and ecological agriculture.

8H2T (IadD/E-IAA). The cryo-EM map of IadD/E in complex with IAA has been deposited to the Electron Microscopy Data Bank under the corresponding accession code EMD-34443. Source data are available in S1 Data.

**Funding:** This work was supported by the National Natural Science Foundation of China (32071218 to H.Z.; 32200496 to G.Y.; 22007072 to H.S. and 32201004 to Z.L.) and the Ministry of Science and Technology, 2022YFA0911800 to Z.L.. The funder had no role in study design, data collection and analysis, decision to publish, or preparation of the manuscript.

**Competing interests:** The authors have declared that no competing interests exist.

**Abbreviations:** ABC, ATP-binding cassette; CTF, contrast transfer function; DAO, dioxygenase for auxin oxidation; FBD, FMN-binding domain; HPLC, high-performance liquid chromatography; HRMS, high-resolution mass spectrometry; IAA, indole-3-acetic acid; IBA, indole-3-butyric acid; IPA, indole-3-propionic acid; IPTG, isopropyl-β-D-thiogalactoside; ITC, isothermal titration calorimetry; MD, molecular dynamics; NAA, 1-naphthylacetic acid; NBD, NADH-binding domain; PAA, phenylacetic acid; PDR, phthalate dioxygenase reductase; RGI, root growth inhibition; SAXS, small-angle X-ray scattering; SBP, solute-binding protein; TSA, thermal shift assay; 2,4-D, 2,4-dichlorophenoxyacetic acid; 4-Cl-IAA, 4-chloroindole-3-acetic acid; 5-HIAA, 5-hydroxy indole-3-acetic acid.

## Introduction

Auxin is a vital phytohormone in plants, affecting almost all the aspects of plant's life. In the meristems, auxin functions as a regulator of cell division, elongation, and differentiation, determining the growth and architectures of shoots and roots [1]. Indole-3-acetic acid (IAA) is a highly abundant, natively synthesized auxin in plants. IAA is not uniformly distributed but instead locally synthesized and transported directionally via the polar distributed PIN and PIN-like transporters in plants [2,3]. The resultant IAA concentration gradients instruct plant development and are therefore dedicatedly regulated in synthesis, inactivation, and degradation [4–6].

Plants are associated with millions of microorganisms. The plant–microbiota interaction serves as an important layer of auxin gradient regulation, tuning plant development, phenotypes, and fitness [7,8]. For instance, most rhizobacteria are found capable of producing IAA [7,9], which works in conjunction with IAA synthesized in plants to stimulate the growth of primary and lateral roots within an optimal concentration range but cause root growth inhibition (RGI) at higher concentrations [7,10]. Apart from the IAA-synthesizing bacteria, some plant-associated bacteria could also degrade IAA [7,11], which could serve as a down-regulating scheme of both bacterially produced and endogenous IAA to fine tune the IAA concentration gradients. Elucidating the underlying mechanism of bacterial IAA regulation would help to maximize the beneficial effects of auxin production or degradation for ecological agriculture [12–14]. Two gene clusters, *iac* and *iaa*, were previously discovered responsible for aerobic and anaerobic degradation of IAA, respectively [15,16].

More recently, a new auxin degradation operon has been reported for strains of *Variovorax*, which actively guides IAA degradation and readily counteracts RGI induced by bacterially produced IAA in plant microbiomes [14], indicating distinguishing root growth promoting traits of the *Variovorax* strains. *Variovorax* is a gram-negative bacteria genus in the family Comamonadaceae, commonly found in the rhizosphere and regulating IAA levels [17]. The auxin IAA degradation (*iad*) operon is found highly conserved among strains of *Variovorax* and unique to the *Variovorax* genus [14,18]. However, transformation of non-IAA-degrading bacteria, such as *Acidovorax* root219, with genomic fragments of the *iad* operon could generate gain-of-function strains, capable of IAA degradation and rescuing RGI [14]. Notably, the *iad* operon shares limited homology with the *iac* and *iaa* gene clusters, representing a new auxin IAA-degradation pathway. Dissecting and defining the functional mechanism of auxin degradation by *Variovorax* is therefore of significant importance for potential applications in ecological and stress agricultures [11].

In this study, we have focused on 10 genes around an overlapped region of 2 genomic fragments of *iad* operon (**Figs 1A and S1A**), which were found sufficient for enabling IAA degradation and/or RGI reversion when transformed to non-IAA-degrading bacteria [14]. Using biochemical and structural approaches, we have elucidated the molecular mechanism of IAA transformation by the *Variovorax* operon. Two proteins, IadK2 andIadD, were found to directly associate with IAA. IadK2, featured with a two-lobed structure, was identified as an ATP-binding cassette (ABC) transporter solute-binding protein (SBP) that binds specifically and strongly with IAA, most likely mediating IAA uptake from the environment. Combined biochemical, structural, and mass spectrometry studies revealed that IAA was transformed to the biologically inactive 2-oxindole-3-acetic acid (oxIAA) by the two-component IadC-IadD/IadE dioxygenase-reductase system encoded by the *iad* operon in *Variovorax*. This is consistent with a latest report published during the preparation of this manuscript [18]. Further, we demonstrated that such an IAA-degradation property of *Variovrax* could be transplanted to *Escherichia coli* by the transformation of a minimum gene set containing *iadC/D/E*. Together,

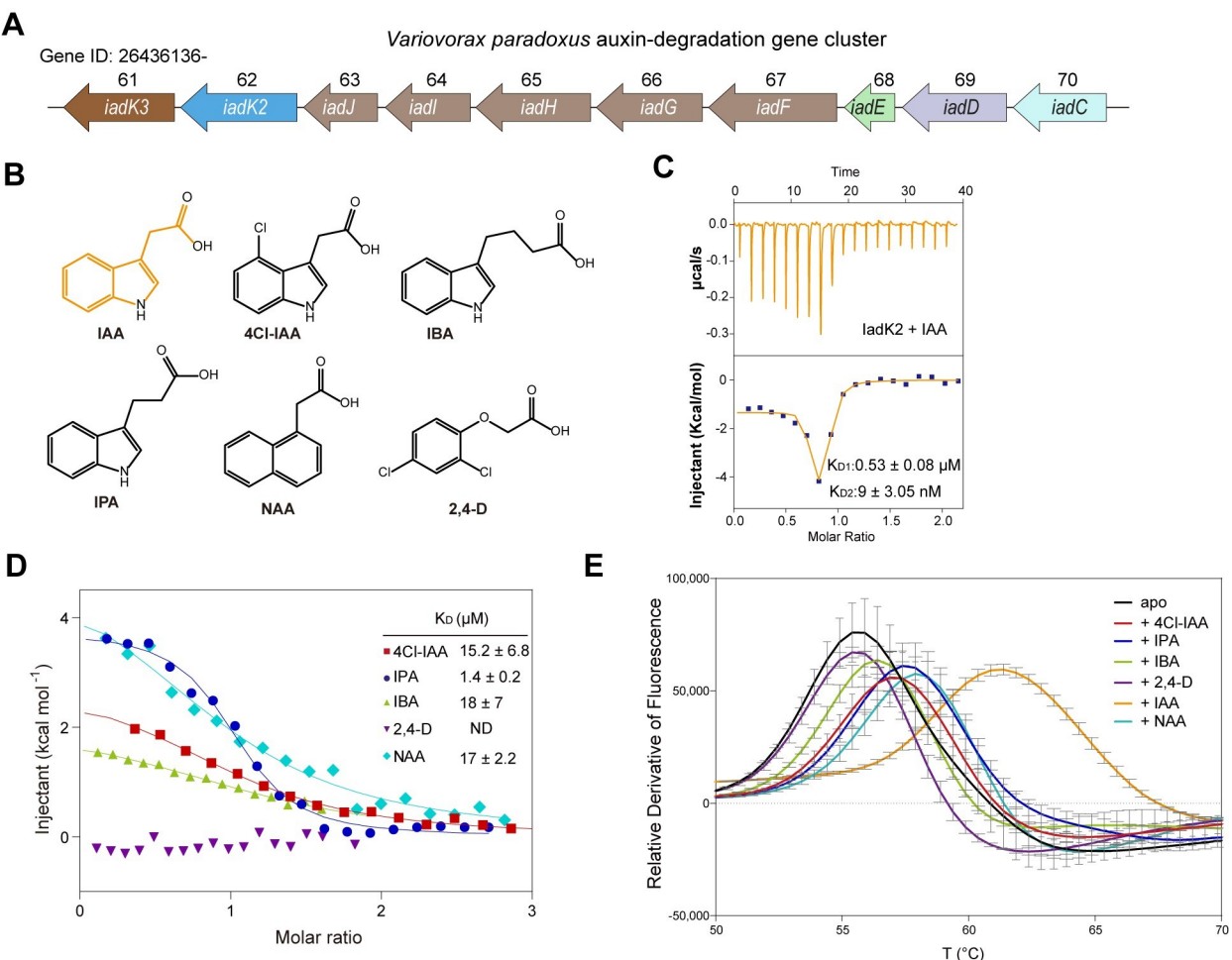

**Fig 1. IadK2 encoded by the *Variovorax iad* operon selectively binds with auxin IAA.** (**A**) Schematics for the auxin IAA-degradation (*iad*) operon in *Variovorax paradoxus* CL14. Ten genes ranging from IMG gene ids 2643613661–2643613670 are illustrated. (**B**) Chemical structures of IAA and auxin analogs. IAA is the most dominant natural auxin in most plants. Other endogenous or synthetic auxins or analogs such as 4-Cl-IAA, IBA, IPA, NAA, and 2,4-D are also displayed. (**C**) ITC measurement of IAA binding to iadk2. The binding isotherm for iadk2-IAA is biphasic, revealing 2 dissociation constants. Three independent measurements were performed and the Kd values are shown as mean ± SEM. (**D**) ITC measurements of auxin analogs to iadk2. The binding of other auxins and analogs to iadk2 is distinct from IAA with a single-phase isotherm and substantially reduced binding affinity. ND: no binding detected. Kd values are shown as mean ± SEM from 3 or more independent measurements. (**E**) TSA for iadk2 with IAA and analogs. The orthogonal TSA experiments were performed for iadk2 with different auxins and analogs to further determine the binding specificity. Three replicates of each TSA experiments were performed. Error bars represent the standard deviations. Source data for **C–E** can be found in **S1 Data**. IAA, indole-3-acetic acid; IBA, indole-3-butyric acid; IPA, indole-3-propionic acid; ITC, isothermal titration calorimetry; NAA, 1-naphthylacetic acid; TSA, thermal shift assay; 4-Cl-IAA, 4-chloroindole-3-acetic acid.

our results uncover the major underlying molecular mechanism of IAA turnover by a new auxin degradation gene cluster in *Variovorax*, providing guidance for potential plant microbiota manipulation and optimization for ecological farming.

## Results

### IadK2 directly associates with IAA

To understand the molecular mechanism of IAA turnover by *Variovorax iad* operon, we first set out to identify the components responsible for IAA binding (**Fig 1A and 1B**). SBPs located in the periplasm work in concert with ABC transporters to transport a variety of compounds

into bacteria [19,20]. The iad operon is predicted to encode 2 SBPs, IadK2 and IadK3 (**Figs 1A and S1A**), which could potentially mediate the uptake of IAA in the rhizosphere. To determine whether IadK2 and IadK3 could bind with IAA, we expressed and purified the proteins for isothermal titration calorimetry (ITC) measurements. IadK2 could engage IAA with nanomolar affinity, whereas no obvious binding was detected for IadK3 and IAA (**Figs 1C and S1B**). To further understand the specificity of IadK2, we also tested the binding of IadK2 with other endogenous and synthetic auxins including 4-chloroindole-3-acetic acid (4-Cl-IAA), indole-3-butyric acid (IBA), indole-3-propionic acid (IPA), 1-naphthylacetic acid (NAA), and 2,4-dichlorophenoxyacetic acid (2,4-D) (**Fig 1B**). IadK2 could bind with most bicyclic analogs but not the single aromatic ring 2,4-D, indicating a preference for bicyclic compounds (**Fig 1D**). To test this hypothesis, we further measured the binding between IadK2 and phenylacetic acid (PAA) composed of a single aromatic ring and found indeed no binding between IadK2 and PAA (**S2A Fig**). Additionally, IadK2 associates with the bicyclic auxins and analogs with substantially reduced affinities in comparison to IAA (**Fig 1D**), suggesting the specificity of IadK2 for IAA. This is further supported by the results of the thermal shift assay (TSA), where IAA displayed the most significant stabilization effect on IadK2 (**Fig 1E**). Interestingly, neither IAA nor the auxin analogs could bind to IadK3 (**S1B and 1C and S2B Figs**), suggesting a distinct substrate specificity for IadK3. Together, these data suggest that IadK2 is an IAA-specific SBP in *Variovorax*. The high affinity of IadK2 for IAA may facilitate efficient IAA uptake by *Variovorax* from the environment [21–23].

## Structural basis for IAA engagement by IadK2

SBPs are classified into 7 clusters (A-F) based on the structural similarities [20]. To further understand the IAA binding by IadK2, we next solved the crystal structure of IadK2 at a resolution of 2.3 Å (**S1 Table**). IadK2 is composed of 2 rigid domains, Domain 1 and Domain 2, connected by a flexible hinge region (**Fig 2A and 2B**). Both domains possess the α/β fold, consisting of a central β-sheet flanked by α-helices (**Fig 2B**). The hinge region comprises 3 linkers (L1–L3) with 2 α-helices in linker L2, resembling SBPs in the subcluster B-III (**S3A–S3C Fig**), which are usually associated with type I ABC transporters for aromatic acids uptake [19,20]. Consistently, Dali search revealed a B-III SBP from *Rhodopseudomonas palustris* in complex with caffeic acid as the closest structural homology of IadK2 [24] (**S3C Fig**).

As co-crystallization or crystal soaking with IAA failed after extensive attempts, we therefore performed the docking and molecular dynamics (MD) simulation analysis to understand the interaction between IAA and IadK2 (**Fig 2C**). An IAA-binding pocket at the interface of domain 1 and domain 2 was predicted. Notably, the computed IAA-bound structure is in a "closed" conformation in comparison to our substrate-free crystal structure (**Fig 2B and 2C**), suggesting IadK2 might switch from a resting "open" conformation to a "closed" conformation upon ligand binding. Small-angle X-ray scattering (SAXS) analysis also supported the "closed" conformation for IAA-bound IadK2 (**S3D Fig**). This is analogous to the ligand-induced "Venus Fly-trap"-like closure observed for other SBPs [25]. Next, mutagenesis studies were performed to validate the predicted pocket. Most tested mutations reduced or almost abolished the binding to IAA (**Fig 2D and 2E**). Particularly, substitutions of Phe168, Trp172, Arg199, Phe308, and Tyr311 significantly compromised IAA binding (**Fig 2D and 2E**). Phe168 and Phe308 are conserved among SBPs in the B-III cluster. Trp172 and Arg199 are moderately conserved but Tyr311 is more variable among B-III SBPs (**S3E Fig**).

IadK3 shares approximately 70% sequence identity with IadK2. To elucidate why IadK3 fails to bind IAA, we predicted the structure using AlphaFold2 [26]. As revealed, IadK3 and

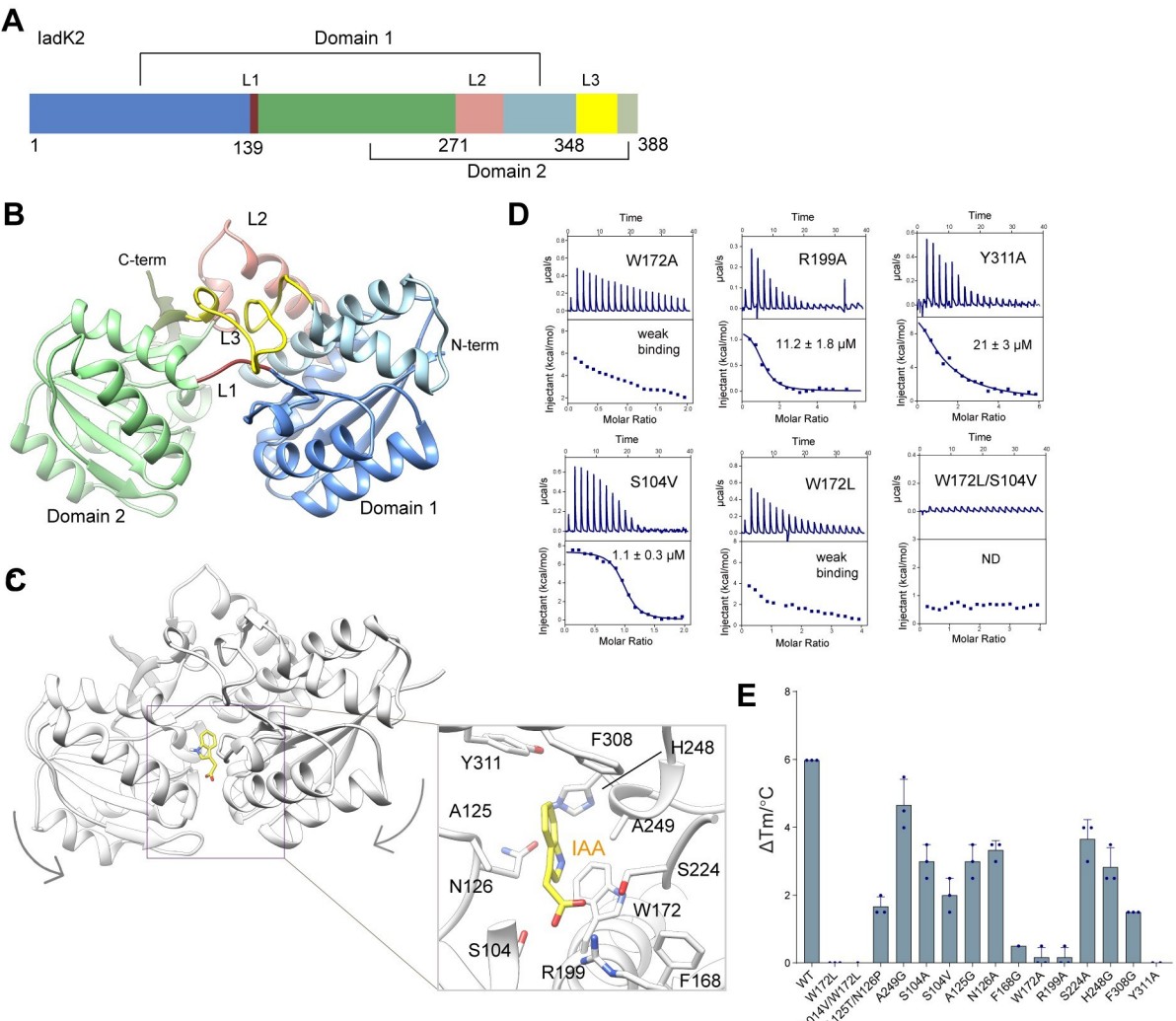

**Fig 2. Structure of IadK2 and the IAA-binding pocket.** (**A**) Domain organization of iadk2. Iadk2 is composed of 2 domains, Domain 1 and Domain 2. (**B**) Crystal structure of iadk2. Iadk2 features a two-lobed structure with Domain1 and Domain 2 connected by a linker region. L1–L3 indicate the 3 linkers connecting the 2 lobes of iadk2. Such overall structural features and that L2 contains 2 α helices group iadk2 to the B-III subcluster of sbps. The same color scheme in **A** is applied. (**C**) Structure of iadk2 complexed with IAA calculated from docking and MD simulation. A similar substrate pocket at the interface of the 2 lobes of iadk2 was predicted, resembling other sbps. The right panel displays the calculated IAA-binding pocket. Iadk2 is colored in gray and IAA is shown as yellow sticks. Key residues potentially involved in IAA binding are shown in sticks representation. (**D**) ITC measurements of IAA binding to iadk2 mutants. ND: no binding detected. Weak binding: KD > 100 µM. Kd values are shown as mean ± SD from 3 or more independent measurements. (**E**) TSA for iadk2 mutants and IAA. Three replications of each TSA experiments were performed. Data are presented as mean ± SEM. Source data for **D and E** can be found in **S1 Data**. IAA, indole-3-acetic acid; ITC, isothermal titration calorimetry; MD, molecular dynamics; TSA, thermal shift assay.

IadK2 share significant structural similarity (**S3B Fig**). Structural comparison revealed several residues located in the binding pocket in IadK2 are not conserved in IadK3, such as Trp172, Ser104, Thr125, and Asn126 in IadK2 (**S3A, S3B and S3E Fig**). Supporting the importance of these residues in defining IAA binding, replacements with corresponding residues in IadK3 obviously impaired the binding of IadK2 with IAA (**Fig 2D and 3E**). Notably, the double mutation replacing both Trp172 and Ser104 with those of IadK3 (W172L/S104V) completely abrogated the binding of IAA. Therefore, Trp172 and its neighboring residues such as Ser104, Thr125, and Asn126 may play important roles in determining IadK2 specificity for IAA.

## Characterization of the IadD/IadE heterocomplex

Apart from IadK2, IadD could also directly bind with IAA, whereas other proteins encoded by the *iad* operon did not (**Figs S1D, S1E, and S4A**). Sequence analysis suggested genes *iadD* and *iadE* may encode the large and small subunits of an aromatic-ring-hydroxylating dioxygenase, respectively. Indeed, IadD and IadE co-migrated in gel filtration assay roughly in a 1:1 molar ratio (**S4B Fig**). Consistently, the IadD/E complex directly binds IAA with a Kd value of about 17 μM (**Fig 3A**).

To further understand the structural features of the IadD/E complex, we determined the crystal structure at a resolution of 1.8 Å (**Fig 3B and S1 Table**). Dali search revealed the biphenyl dioxygenase from *Burkholderia xenovorans* LB400 as the closest homolog [27], which is a Rieske non-heme iron dioxygenase. The Rieske non-heme iron dioxygenases, typically composed of α and β subunits that are equivalent to IadD and IadE, respectively, share a conserved α3β3 configuration [28]. Data from the analytical ultracentrifugation analysis IadD/E also supported a heterohexamer model in solution (**S4C Fig**), indicating IadD/E may share the same α3β3 configuration.

IadD features a Rieske domain (residues 44–160) and a catalytic domain (residues 1–43 and 161–437) (**Fig 3B**). The catalytic domain contains a core helix-grip fold with an eight-stranded antiparallel β-sheet surrounded by 10 helices. The Rieske domain folds into 2 parallel β-sheets composed of 3 and 4 β-strands, respectively, which are connected by multiple long loops. IadE is a single-domain protein, containing a twisted seven-stranded β-sheet and 3 α-helices, which may mainly serve as a scaffold protein contributing to proper assembly of the oligomeric enzyme complex.

To understand the binding of IAA with IadD, we then determined a 2.6 Å cryo-EM structure of IadD/E complexed with IAA (**S7 Fig and S2 Table**). An overall mushroom-shaped heterohexamer composed of 3 IadE at the "stem" and 3 IadD in the "cap" was resolved (**Fig 3C**). The active site of IadD is located at the interface of IadD subunits, composed of a Rieske [2Fe-2S] cluster from the Rieske domain and a mononuclear iron in the catalytic domain of a neighboring subunit (**Figs 3D and S5A and S5B**). The [2Fe-2S] is stabilized by contacts with Cys85, His87, Cys106, and His109 and the mononuclear iron coordinates with His216, His221, and Asp377. Adjacent to the mononuclear iron is the IAA-binding pocket lined by polar and hydrophobic residues including Asn210, Leu211, His216, His221, Phe251, His311, Lys322, Arg329, and Tyr362 (**Figs 3E and S5A and S5B**). No profound overall conformational changes were observed for IadD/E after binding of IAA (**Figs 3D and S5C**). The side chain of His311 in IadD is relocated to accommodate the binding of IAA (**S5D Fig**). The mononuclear iron and the coordinating residues also move slightly inwards (about 1.6 Å) following the binding of IAA. The resolved IAA-binding pocket of IadD/E is largely superimposed with that of biphenyl in the biphenyl dioxygenase structure, but contains different residues to accommodate the substrate specificity (**S6A Fig**). As anticipated, mutation of residues in the pocket abrogated IAA binding (**Fig 3F**). These results therefore suggest IadD/E is a Rieske non-heme dioxygenase, which may use IAA as a substrate.

## IadD/IadE and IadC reconstitute a two-component dioxygenase system specific for IAA transformation

The Rieske dioxygenases usually work in combination with a reductase or both reductase and ferredoxin components, forming a two- or three-component dioxygenase system, to oxidize substrates [28]. Indeed, analysis of the *iad* operon revealed the neighbor gene *iadC* that encodes a reductase protein (IadC) (**Figs 1A and S1A**). To test if IadC could work together with IadD/E to transform IAA, we purified the IadC protein and performed the in vitro IAA

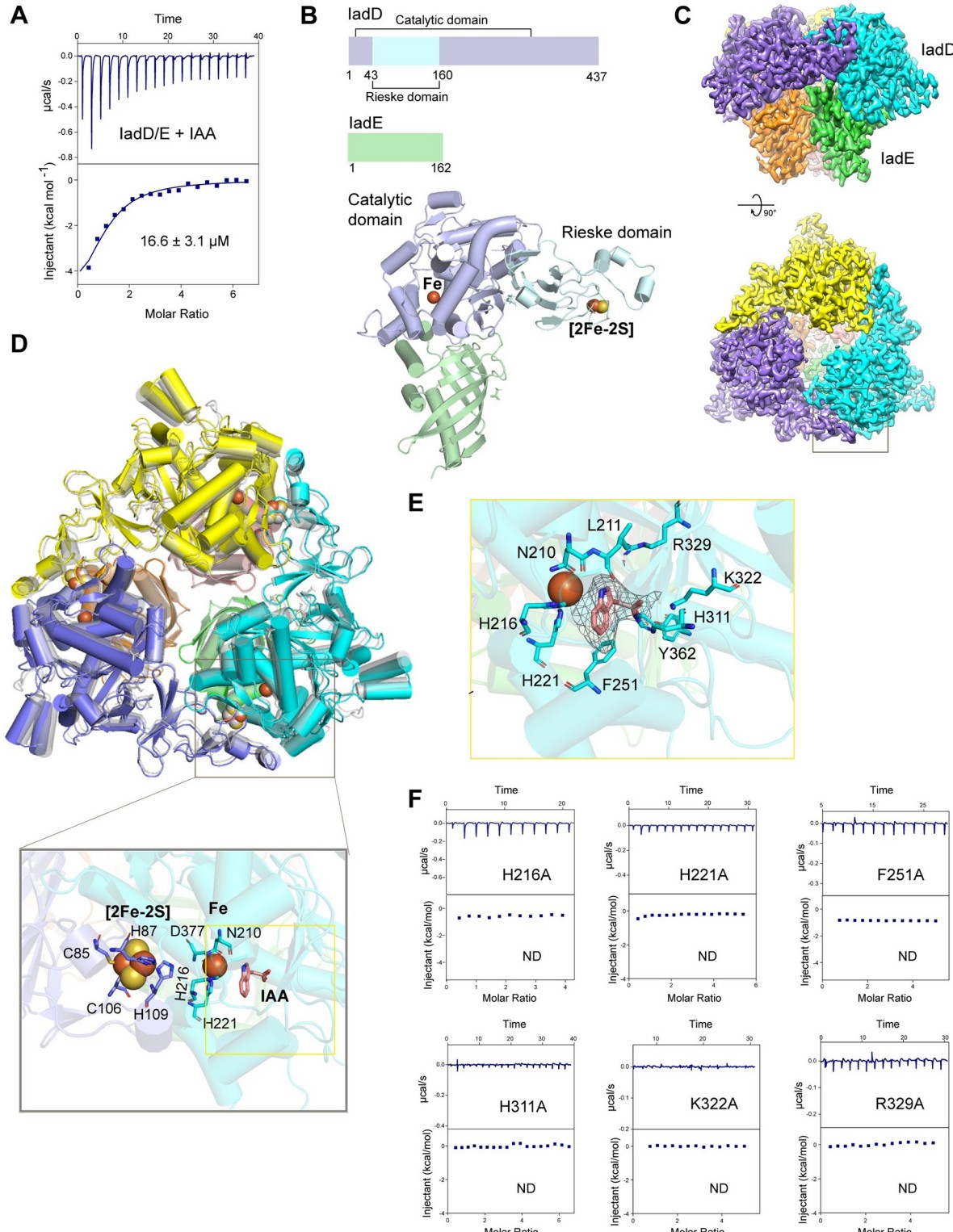

**Fig 3. Characterization of the IadD/Enon-heme Rieske dioxygenase complex.** (**A**) ITC measurement of IAA binding to the E complex. Kd values are shown as mean ± SEM from 3 or more independent measurements. (**B**) Crystal structure of iadd/E. Upper panel: Domain organization of iade and iadd. Bottom: one heterodimer of iadd and iade in the asymmetric unit. Iadd is composed of 2 domains. The catalytic domain of iadd is colored in light blue and the Rieske domain is colored in pale cyan. Positions of the ferrous iron in the catalytic domain and the [2Fe-2S] cluster in the Rieske domain are indicated. The iade is a single domain protein and is colored in pale green. (**C**)

The 2.6 Å cryo-EM density of the iadd/E complexed with IAA. A mushroom-shaped, heterohexameric structure was resolved. The top and side views are displayed. (**D**) Overlay of the apo and IAA-bound structures of iadd/E. The IAA-bound heterohexamer of iadd/E (colored by subunits) was superimposed with that of the apo state generated with crystal symmetry-equivalent copies (colored in gray). No significant conformational changes were identified except some regions at the crystal-packing interfaces. The inset panel shows a zoom-in view of the active site composed of the [2Fe-2S] cluster, the mononuclear iron and the substrate-binding pocket. (**E**) The IAA-binding pocket in iadd. The indole ring of IAA and the mononuclear iron are in relatively perpendicular orientation. Key residues in the IAA-binding pocket are shown as sticks. (**F**) ITC measurements of IAA binding to iadd mutants. Mutations of residues in the IAA-binding pocket as shown in **E** disrupted the binding. ND: no binding detected. Source data for **A and F** can be found in **S1 Data**. IAA, indole-3-acetic acid; ITC, isothermal titration calorimetry.

transformation experiment. Indeed, IAA was transformed efficiently in the presence of both IadD/E and IadC, whereas neither IadD/E nor IadC alone could catalyze the transformation (**Fig 4A and 4B**), suggesting a potential two-component dioxygenase system for IAA conversion.

Next, we determined the crystallographic structure of IadC at a resolution of 1.68 Å (**S1 Table**). The overall triangle-shaped protein is composed of 3 major domains, including an FMN-binding domain (FBD, residues 1–115), an NADH-binding domain (NBD, residues 116–238), and a plant-type [2Fe-2S] cluster domain (FeSD, residues 239–329) (**Fig 4C**). The FBD folds into a six-stranded β-barrel and the NBD features an α/β fold, forming a five-stranded β-sheet wrapped by helices. FMN resides at the interface of the FBD and NBD domains, with the isoalloxazine ring sandwiched between the side chains of His59 and Phe233 and anchored by polar contacts with nearby residues. Further, the ribityl moieties of FMN are stabilized by polar contacts with residues Arg58, Arg89, and Ser92 (**Fig 4C**, right panel). FeSD is composed of a twisted β-sheet with an α-helix positioned at each side. The [2Fe-2S] cluster is coordinated by 4 Cys residues in 2 peripheral loops, facing the center of the tri-lobbed IadC (**Fig 4C**, bottom panel). IadC closely resembles the phthalate dioxygenase reductase (PDR) from *Pseudomonas cepacia* according to Dali analysis [29] (**S6B Fig**), indicating a classical IA FMN-type reductase. However, the relative domain orientations are distinct between the 2 structures. While FBD and NBD are closely packed in both PDR and IadC structures, FeSD associates primarily with NBD in IadC but with FBD in the PDR (**S6C Fig**), potentially representing the different states of electron transfer [28].

Mutagenesis studies were then performed to further confirm the functional role of IadC in IAA transformation. FMN is thought to mediate electron transfer from the NADH electron donor to the [2Fe-2S] electron acceptor, which further delivers the electrons to the dioxygenase for the catalysis reaction [29,30] (**Fig 4C and 4E**). As anticipated, mutation of residues involved in FMN binding compromised IAA turnover (**Fig 4D**). Similarly, the C279S mutation impairing coordination of the [2Fe-2S] cluster significantly decreased the efficiency of the reaction. Likewise, mutation of residues involved in electron transfer and IAA binding in IadD also significantly impaired the processing of IAA (**Figs 3E and 4E and 4F**), supporting the critical role of the catalytic activity of IadD in IAA transformation.

In addition to IAA, other auxin analogs also play essential roles in plant growth and development. We therefore further examined whether the IadD/E-IadC system could transform IAA analogs. While 4-Cl-IAA was transformed with reduced efficiency, IadD/E-IadC could not catalyze the turnover of IAA analogs such as IBA and IPA (**S8A and S8B Fig**). Thus, the IadD/E-IadC dioxygenase system appears to be selective for the transformation of IAA.

## Metabolite identification of IAA degraded by the IadD/E-IadC dioxygenase system

Next, we set out to identify the product of IAA transformed by this new dioxygenase system. To this end, we performed the high-performance liquid chromatography (HPLC) and the

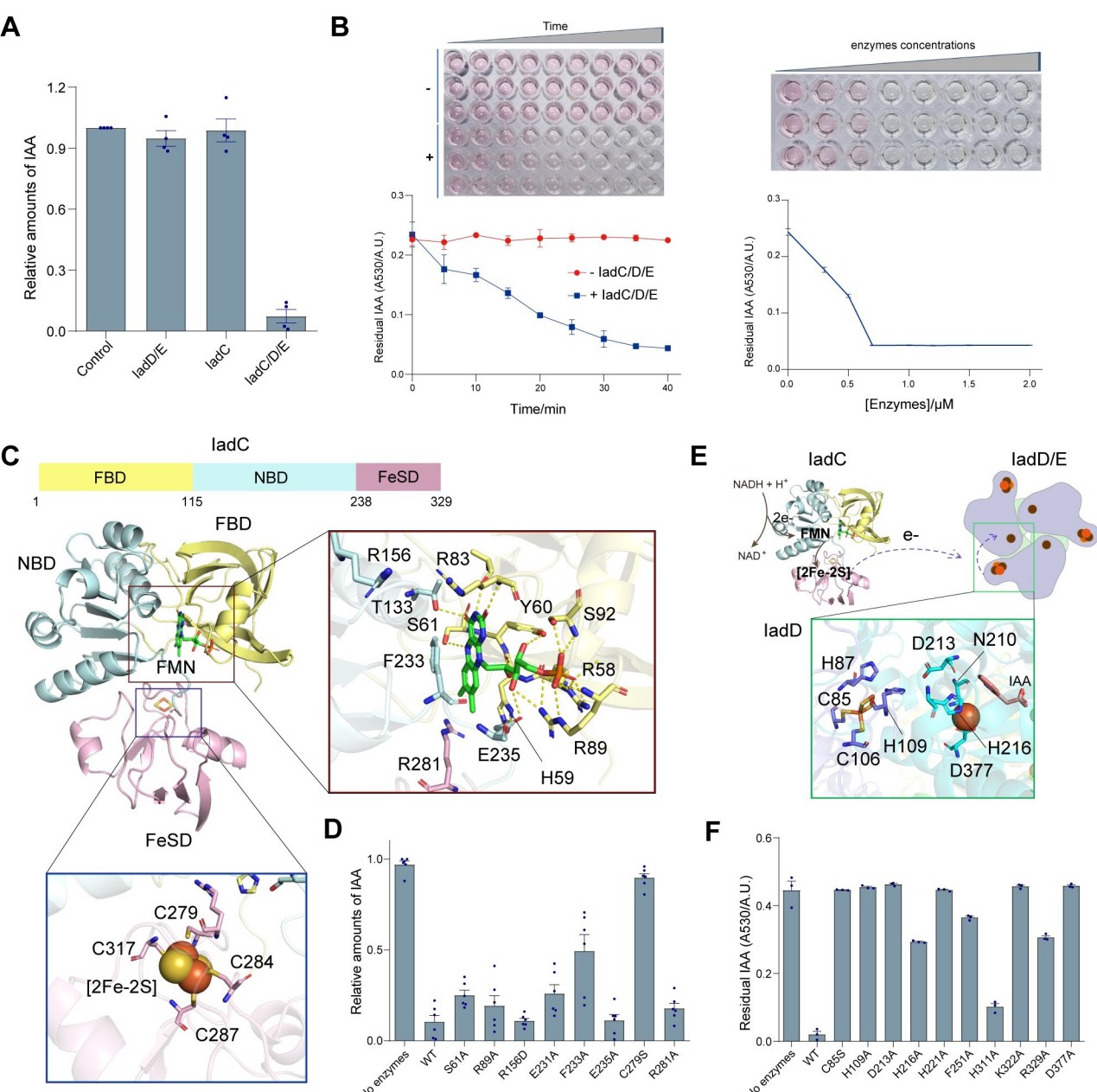

**Fig 4. Reductase IadC and dioxygenase IadD/E catalyze IAA transformation.** (**A**) IAA transformation assay. Purified iadd/E and iadc were used to test the in vitro IAA transformation. The amounts of IAA were quantified using the Salkowski reagent and adsorption at 530 nm. Data are presented as mean ± SEM. (**B**) Time and concentration dependence of IAA transformation by iadc and iadd/E. IAA mixed with the Salkowski reagent show pink color, which turned colorless after treatment with iadc and iadd/E. Top panels show the color change of IAA reaction mixtures in the presence of Salkowski reagent over time (left, 0.7 μM iadc/D/E enzymes) or concentrations of iadc/D/E (right, reacted for 30 min), from pink to colorless. Below graphs illustrate the changes of quantified IAA amounts (adsorption at 530 nm) over time (left) or enzyme concentrations (right). (**C**) The determined crystal structure of iadc. Iadc possesses a tri-lobed structure, consisting of 3 domains including FBD, NBD, and fesd. The [2FE-2S]-binding pocket in fesd and the FMN-binding pocket in FBD are displayed, with key residues participated in the binding indicated. (**D**) Effects of iadc mutations on IAA transformation. In vitro IAA transformation assay was performed with both WT and mutated iadc in the presence of iadd/E. Data are presented as mean ± SEM. (**E**) Schematic illustration of the functional cooperation of iadc and the iadd/E. Iadc transfers electron to the active site of iadd composed of [2Fe-2S] and ferrous iron to catalyze IAA transformation. The inset panel displays key residues in iadd surrounding the [2Fe-2S] cluster and the ferrous iron, potentially essential for the electron transfer. (**F**) Effects of iadd mutations on IAA transformation. Both WT iadd/E and mutants defective in substrate binding or electron transfer were tested with the in vitro IAA degradation assay. Data are presented as mean ± SEM. Source data for **A, B, D,** and **F** can be found in **S1 Data**. FBD, FMN-binding domain; IAA, indole-3-acetic acid; NBD, NADH-binding domain; WT, wild type.

high-resolution mass spectrometry (HRMS) analyses. IAA was treated in vitro with the IadD/E-IadC enzymes till complete consumption of IAA. The reacted mixture was first denatured and analyzed using HPLC. A new peak with a retention time of 5.4 min was observed accompanied with the disappearance of the IAA peak (**Figs 5A** and **S9**). Fractions from this potential product peak were then collected and subjected to the HRMS analysis (**Fig 5B**). Two possible products were proposed based on the HRMS spectra, including the 5-hydroxy indole-3-acetic acid (5-HIAA) and the oxIAA. IAA develops the pink color with the Salkowski reagent [31], which turned colorless after transformation by IadD/E-IadC (**Fig 4B**), indicating a product unreactive with Salkowski reagent. It therefore ruled out 5-HIAA as the potential product as that a dark gray color was observed after mixing 5-HIAA with the Salkowski reagent and 5-HIAA could be further degraded by IadD/E-IadC (**Figs 5C** and **S8**). By contrast, oxIAA was found unable to react with the Salkowski reagent (**Fig 5C**). Furthermore, the product peak overlapped with that of oxIAA and the peak intensity increased following the addition of oxIAA to the reacted mixture in the HPLC assays (**S9 Fig**); the adsorption spectrum of the reacted mixture also overlaid with that of oxIAA (**Fig 5D**), both supporting oxIAA as the product.

Domains of IadD/E and IadC would reconstitute the electron transport chain of Rieske dioxygenase, where electrons from NADH in NAD are passed to the [2Fe-2S] cluster by FMN bound to FAD of IadC followed by transporting to the active site of IadD through the [2Fe-2S] clusters of IadD/E, catalyzing the incorporation of oxygen atoms of $O_2$ into IAA [28–30,32]. Indeed, in the isotopic experiment with 95% $H_2O^{18}$ and $^{16}O_2$, we found most oxIAA with m/z = 192.0655 containing $^{16}O$-bearing amide, indicating the oxygen was from $^{16}O_2$ (**S10 Fig**). Together, these data suggest that IAA was converted to oxIAA by the dioxygenase system composed of the IadD/E dioxygenase and the IadC reductase (**Fig 5E**). Interestingly, a recent study also reported the catalyzed conversion of IAA to 2-hydroxyindole-3-acetic acid intermediate by the *iad* operon in bacteria [18], which is chemically unstable and would form the stable "keto" form, oxIAA, as shown in our study.

## Engineering of *E. coli* for IAA transformation

We next explored the possibility of engineering non-IAA-degrading bacterial strains for IAA transformation. Vectors containing different combinations of genes for *iadK2*, *iadD/E*, and *iadC* were generated and transformed to *E. coli* to evaluate the IAA conversion capabilities (**Fig 6A**). Indeed, we found transformation of the *iad* genes to *E. coli* could enable bacterial IAA conversion. Consistent with the in vitro assay, *iadC*, *iadD*, and *iadE* were required for IAA transformation by *E. coli*. Nevertheless, *IadK2* appeared to be dispensable, as *E. coli* could efficiently transform IAA regardless of the presence of *IadK2* under the experimental condition (**Fig 6B**), indicating the potential presence of endogenous IAA transporters in *E. coli*. Consistently, the loss-of-function mutations in IadD or IadC significantly compromised or abrogated IAA conversion by the engineered *E. coli*, whereas mutation of IadK2 showed no obvious effect on IAA transformation (**Figs 6C** and **S11**). These results therefore suggest that incorporation of a minimum set of genes containing *IadE*, *IadD*, and *IadC* to *E. coli* could enable bacterial transformation and inactivation of IAA.

## Discussion

Bacterial degradation of plant hormone plays an important role in regulating plant hormone levels [8,16]. Recently, a conserved *iad* operon coding for a new auxin IAA degradation pathway was discovered among strains in the genus *Variovorax*, which is responsible for reversing the RGI and promoting plant growth [14]. Here, we identified and characterized the proteins

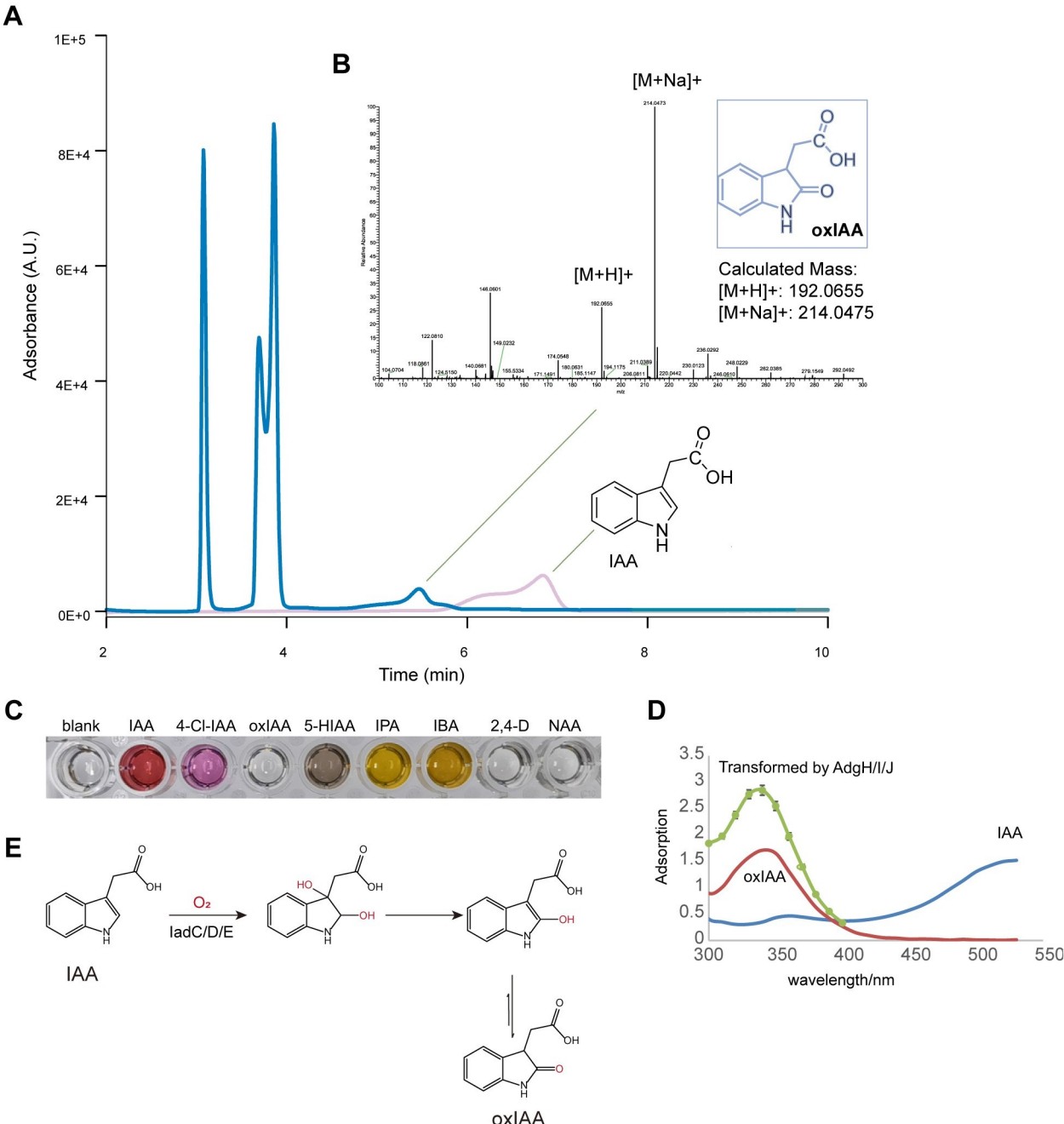

**Fig 5. The IadD/E-IadCdioxygenase-reductase system converted IAA to oxIAA.** (**A**) HPLC analysis of IAA before (in pink) and after transformation (in blue) by iadd/E-iadc. (**B**) HRMS analysis of the transformed products of IAA by iadd/E-iadc. The elution for potential product peak in HPLC was collected and analyzed by HRMS. Characteristic peaks for oxiaa were identified, which agreed with the calculated mass. Source data for **A** and **B** can be found in **S1 Data**. (**C**) Color development of IAA and analogs in Salkowski reagent. The listed compounds (0.5 mm) developed different colors after mixing with the Salkowski reagent. (**D**) The absorbance spectra of IAA, oxiaa, and the products by iadd/E-iadc. The spectrum scan for IAA, oxiaa, and transformed IAA by the iadd/E-iadcafter mixing with the Salkowski reagent. (**E**) The mechanism of IAA transformation by iadd/E-iadc. The iadd/E-iadcdioxygenase system catalyzes the incorporation of oxygen atoms to IAA from $O_2$, leading to the formation of 2-hydroxyindole-3-acetic acid, which potentially primarily exist as the stable "keto" form, oxiaa. HPLC, high-performance liquid chromatography; HRMS, high-resolution mass spectrometry; IAA, indole-3-acetic acid.

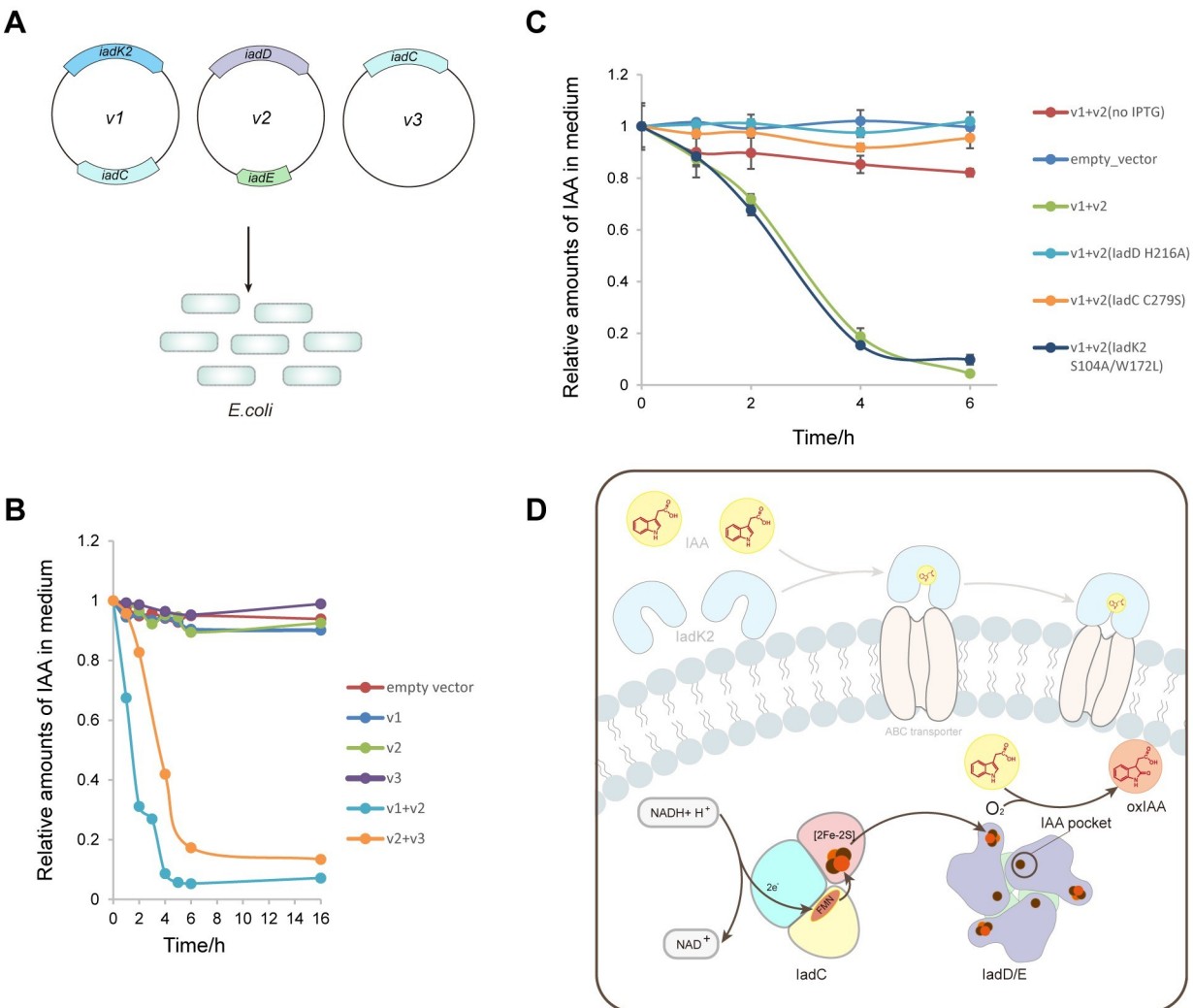

**Fig 6. Ectopic expression of *Variovorax* genes *iadC/D/E* enables bacterial transformation of IAA by *E. coli*.** (**A**) Schematic for the design of the experiment for the bacterial IAA transformation assay. Vectors containing the indicated *iad* genes were transformed to *E. Coli* and the transformation of IAA in the LB medium was monitored using the Salkowski reagent. (**B**) In vivo IAA transformation assay for *E. coli* containing different *iad* genes as indicated in **A**. Little IAA transformation was observed when the empty vector carrying no *iad* genes was transformed to *E. coli*. The presence of all 3 genes *iadc/D/E* resulted in efficient IAA transformation. (**C**) Effects of Iad proteins mutants in the in vivo IAA transformation by *E. coli*. Loss-of-function mutations of iadk2 (S104V/W172L), iadd (H216A), and iadc (C279S) were tested. Minimal IAA transformation occurred with the empty vector or in the absence of protein expression inducer IPTG. Source data for **B and C** can be found in **S1 Data**. (**D**) Schematic diagram of IAA transformation by the *Variovorax iad* operon. IAA transported to the bacterial cell would be first processed by the iadd/E-iadc dioxygenase system and converted to oxiaa in the presence of molecular oxygen, thereby deactivating the auxin IAA. Despite not required for the engineered *E. coli*, iadk2 may mediate the efficient IAA uptake by *Variovorax* from the environment. However, further studies would be required to examine whether iadk2 is need for *Variovorax* IAA degradation. IAA, indole-3-acetic acid; IPTG, isopropyl-β-D-thiogalactoside.

produced by the *iad* operon, directly involved in the processing of auxin IAA. Our study suggests IAA could be transformed to oxIAA by an Rieske non-heme dioxygenase system constituted by IadD/E and IadC encoded by the *iad* operon (**Fig 6D**). As oxIAA is biologically inactive [4], products of the *iadC/D/E* genes are likely sufficient for the bacterial down-regulation of IAA in the rhizosphere to reverse RGI. Indeed, a latest study published during the preparation of this manuscript also reveals the same minimum gene set for IAA inactivation and the reversion of RGI [18]. Interestingly, gene *iadC* was found strictly required in some

bacterial strains but dispensable in others, indicating that the function of IadC may be (partially) compensated by other reductase proteins in the latter. Notably, the IadD/E-IadC catalyzed IAA conversion to oxIAA is reminiscent of IAA inactivation in plants by the plant protein dioxygenase for auxin oxidation (DAO) [33], which is the dominant catabolic pathway in plant for IAA concentration regulation.

In addition to the IadD/E-IadC proteins, we also found IadK2 may also contribute to the IAA-degradation pathway of *Variovorax* by facilitating IAA transportation, although it was found dispensable for *E. coli* (**Fig 6B**). The *iad* operon encodes 2 SBPs that are located between the outer and inner membranes of gram-negative bacteria and typically required for nutrient and signaling molecules uptake by the ABC transporters [34], IadK3 and IadK2. Both IadK3 and IadK2 belong to the B-III subclass of SBPs with a binding specificity for aromatic compounds [20]. Despite the significant structural and sequence similarities, the 2 SBP proteins have different ligand specificity. While IadK3 show no detectable binding, IadK2 binds to IAA with nanomolar affinity. Considering the high binding affinity and the micromolar concentrations of IAA in the environment [21–23], IadK2 may serve as a highly efficient IAA "grasper," working with ABC transporter in the membrane and mediating IAA uptake in *Variovorax*. This property may contribute to the supreme capability of *Variovorax* strains in degrading IAA and reverse RGI [14,17]. IadK2 also binds with other auxins with structural similarities such as 4Cl-IAA, NAA, IPA, and IBA, despite the substantially reduced affinity (μM or lower). IadK3 however show no binding with these auxin analogs. Interestingly, we found IadK3 could weakly associate with oxIAA. The detailed roles of IadK3 and IadK2 in the *Variovorax* auxin degradation pathway, however, remain to be further characterized and defined.

IadD and IadE form a heterohexameric non-heme Rieske dioxygenase. IadC is an FMN-type reductase, which transfers electrons from the donor NADH to the active site of IadD to catalyze the incorporation of oxygen atoms into IAA for the production of oxIAA. This may represent the first step in the bacterial degradation of IAA by the *iad* operon. The IadD/E-IadC dioxygenase system appears to be specific for IAA (**S8 Fig**), similar to the ligand specificity of IadK2, indicating the preference for IAA for the *iad* operon pathway. The biologically inactive oxIAA is a major form of the irreversible oxidation in plants regulating IAA levels [5]. The IadD/E-IadC-catalyzed conversion of IAA to oxIAA is therefore likely the most essential step in IAA degradation and RGI reversion of plants. Recently, the 2-hydroxyindole-3-acetic acid was reported as the first-step intermediate in the metabolite analysis of *Variovorax paradoxus* CL14 cultured in the presence of IAA [18]. It should be noted that the 2-hydroxyindole-3-acetic acid as an "enol" is chemically unstable, which would be converted into the stable "keto" form, oxIAA (the major tautomer) if no further degradation occurs as shown in our study. Inside bacteria, the intermediates would likely be further degraded to anthranilic acid in the *iad* pathway [18]. The elucidated molecular mechanism underlying the most essential step in IAA degradation by the *iad* operon, could be translated for applications in engineering and optimizing the rhizosphere microbiome [35,36]. For example, we have demonstrated that incorporation of a minimum gene set containing *iadC/D/E* could enable *E. coli* to degrade IAA. Further studies are required to elucidate the functional roles of other components in this pathway and the underlying mechanism of the following decomposition.

## Methods

### Protein expression and purification

DNA sequences for genes in *Variovorax paradoxus* CL14 *iad* auxin degradation operon were subcloned into expression vectors for protein purification. Genes encoding IadK3 (IMG gene ID: 2643613661), IadK2 (IMG gene ID: 2643613662), and IadG (IMG gene ID: 2643613666)

were inserted into pET28-SUMO vector, and sequences for IadH-J (IMG gene IDs: 2643613663–2643613665), IadE-F (IMG gene ID: 2643613667, 2643613668), and IadC (IMG gene ID: 2643613670) were cloned into pET28-MHL vector. Two constructs for IadD (IMG gene ID: 2643613669) were generated by subcloning to vectors pET28-SUMO and 13S-A (Addgene, 48323) vectors, respectively. The resultant plasmids were transformed into Rosetta DE3 cells for protein expression. IadE and IadD were co-expressed to obtain the functional complex. Protein expression was induced with isopropyl-β-D-thiogalactoside (IPTG) at OD600 of 0.6 and 37˚(. Cell pellets were resuspended in the binding buffer (25 mM Tris-HCl (pH 7.5), 500 mM NaCl, 5 mM imidazole, and 2 mM β-mercaptoethanol) and broken by sonication. Ni-NTA resins (Qiagen) were added and incubated with the cleared lysate at 4˚ for 1 h before being extensively washed with the washing buffer (binding buffer supplemented with 15 mM imidazole). Target protein was eluted with the elution buffer containing 300 mM imidazole. The expression tags were removed by TEV protease. Tag-free proteins were concentrated and loaded onto the HiTrap SP column (Cytiva) for further purification. Gel-filtration was performed at the last step of purification on a ÄKTA Pure system with running buffer containing 25 mM Tris-HCl (pH 7.5), 150 mM NaCl, 2 mM dithiothreitol (DTT). Peak fractions were collected and concentrated for use. Mutant proteins were purified in the same way as described above.

### Isothermal titration calorimetry (ITC)

ITC experiments were performed using a MicroCal PEAQ-ITC system (Malvern). All the experiments were performed at 25˚P in a buffer containing 25 mM Tris-HCl (pH 7.5) and 150 mM NaCl (unless otherwise stated). A typical titration experiment involved 18 injections of IAA solution (500 to 2,000 μM) into the cell containing protein of interest (30 to 100 μM) with a time spacing of 120 s. Data was analyzed using the MicroCal PEAQ-ITC analysis software. All the measurements were repeated at least 2 times.

### Thermal shift assay (TSA)

TSA was performed to examine the binding of Iad proteins with IAA and analogs. Ligands concentrations were tested at 2- to 100-fold excess (10 to 1,000 μM) relative to the proteins (5 μM). The fluorescence dye SYPRO Orange at a final of 200-fold dilution was used to monitor the unfolding of proteins. The protein and ligand were incubated for 10 min at room temperature before the addition of SYPRO Orange dye. The QuantStudio 1 Plus real-time PCR system (Thermo Fisher) was used to control the temperature increasing from 25˚1 to 99˚t with an increment of 0.5˚n and to monitor the fluorescence (excitation: 520 ± 10 nm and emission: 558 ± 10 nm). Each tested group was run in triplicate and repeated. Data was imported to the Protein Thermal Shift software for analysis and plotting.

### Gel-filtration assay

The gel-filtration assay for the IadD/IadE complex was carried out using the HiLoad 200 column (Cytiva). IadD/E purified by the cation exchange chromatography was concentrated to 20 mg/ml and loaded into the column pre-equilibrated with the running buffer containing 25 mM Tris-HCl (pH 7.5), 150 mM NaCl, and 2 mM DTT. The peak fractions were collected and analyzed with SDS-PAGE.

### Analytical ultracentrifugation analysis

The purified IadD/E complex sample was diluted to 0.5 mg/ml using the sample buffer containing 25 mM Tris-HCl (pH 7.5) and 150 mM NaCl and subjected to the sedimentation

velocity measurements using an Optima XL-I analytical ultracentrifuge (Beckman-Coulter). Data was analyzed with the SEDFIT and SEDPHAT programs [37,38].

## Small-angle X-ray scattering

The IadK2-IAA complex was concentrated to 20 mg/ml in a buffer containing 25 mM Tris-HCl (pH 7.5), 280 mM NaCl, and 2 mM DTT. SAXS data was collected on beamline BL19U2 at Shanghai Synchrotron Radiation Facility (SSRF) with a Pilatus detector. One-dimensional intensity curves were obtained by the ATSAS package [39,40]. SAXS data processing and analysis were carried out using PRIMUS software within the ATSAS package [41]. Distance distribution and molecular weight were obtained from GNOM [42]. Ab initio modeling and low-resolution 3D shape envelopes reconstruction were performed by DAMMIF [43]. CRYSOL and SREFLEX were used to test the fit quality of reconstructed models with the experimental SAXS profiles [40].

## In vitro IAA and analogs transformation assay

In vitro assay was performed to test the capability of IadC/D/E enzymes encoded by the *iad* operon in transforming IAA and analogs. The reaction mixture was composed of 100 mM Tris-HCl (pH7.5), 250 μM reduced NADH, 10 μM ammonium iron (II) sulfate, and 3 μM of each enzyme component. The reaction was initiated by the addition of IAA substrate (0.1 to 0.5 mM) at room temperature (unless otherwise stated). The mixture was vortexed periodically to introduce oxygen for the reaction and was boiled to stop the reaction. After pelleting the precipitants, the supernatant was collected and mixed with 2 volumes of Salkowski reagent made of 0.5 M ferric chloride and 35% perchloric acid for the quantification of residual substrate by measuring the adsorption at 530 nm.

## HPLC and HRMS analysis of IAA metabolite

The reacted mixture with almost all the IAA substrate consumed from the in vitro IAA transformation assay was denatured by heat and the supernatant was collected for HPLC analysis on a Shimadzu LC-40 HPLC machine equipped with an InertSustain 5 μM C18 HPLC column (4.6 × 250 mm). The UV detection was carried out at 260 nm, following separation using water (solvent A) and methanol (solvent B) as mobile phase with a linear gradient elution at a flow rate of 0.8 ml/min as follows: 0 to 5 min, 5% B; 5 to 15 min, 10% B; 15 to 20 min, 50% B; 20 to 22 min, 50% B; 22 to 25 min, 95% B; 25 to 33 min, 95% B. The peak fraction with a retention time of about 5.4 min was collected and re-dissolved in water containing 10% methanol after removing the solvent under vacuum. HRMS was performed with an LTQ-Orbitrap mass spectrometer (Thermo Fisher). Pure IAA, NADH, NAD$^+$, and oxIAA were also analyzed with HPLC for comparison and identification of peaks. The isotopic experiment was carried out in the solution containing 95% $H_2O^{18}$ as well as $^{16}O_2$ in the air following the same protocol as described above.

## IAA bacterial transformation assay

*Variovorax iad* genes *IadK2*, *iadC/D/E* were transformed to *E. coli* BL21(DE3) to screen for the minimum set required for IAA transformation. The *E. coli* cells were cultured in the LB medium till OD600 of 0.6 at 37˚3 and IPTG was then added to induce Iad proteins expression. Two h later, 0.2 mM IAA was added to the medium and the cells were grown for 16 h, during which culture aliquots were collected and the amounts of IAA were determined using the Salkowski reagent and the adsorption at 530 nm.

## Crystallization, data collection, and structure determination

Purified adK2, IadD/E complex, and IadC were concentrated to 15 to 20 mg/ml. Protein crystallization was performed by mixing 1 μl of protein solution with 1 μl of the crystallization buffer using the sitting drop method. Crystals of IadK2 protein were grown in a reservoir solution containing 27% w/v polyethylene glycol 3,350, 0.1 M Tris (pH 8.60); 1 M cobalt (II) chloride hexahydrate. IadD/E crystals were obtained at 18˚E in a reservoir solution containing 28% (v/v) 2-methyl-2,4-pentanediol, 0.05 M sodium cacodylate trihydrate (pH 6.0), 0.05 M magnesium acetate tetrahydrate, 40% (v/v) 1,3-propanediol. Crystals of IadC protein were grown at 18˚p in a reservoir solution containing 0.1 M Bis-tris propane (pH 7.1), 1.2 M sodium malonate (pH 7.0), 0.1 M calcium chloride dihydrate. Crystals were cryo-protected in the mother liquor supplemented with 20% glycerol before flash-frozen in liquid nitrogen for data collection.

X-ray diffraction datasets were collected at beamline BL19U1 of Shanghai Synchrotron Radiation Facility (SSRF). Structure determination was performed by CCP4I2 package [44]. Structure of IadK2 was determined by molecular replacement using a predicted model from Alphafold2 [26]. Structure of the IadD/E protein complex was solved by molecular replacement using the crystal structure of toluene 2,3-dioxygenase (PDB ID:3EN1). Initial model improvement was performed manually in COOT [45]. Model refinement was performed by phenix.refine [46,47] and CCP4I2 package [44]. Data processing and structural refinement statistics are shown in the **S1 Table**.

## Docking and molecular dynamics (MD) simulation

Docking and MD simulation analysis were performed to define the binding site of IAA in IadK2. The initial binding pocket was selected by referring to the homologous structure (PDB ID: 4FB4). Molecular docking of IAA anion to the potential binding site was performed using AutoDock Vina [48]. The revealed initial binding pose served as the start point for MD simulation. MD simulations were performed using the OpenMM and Ambertools software packages [49,50]. The anion IAA was modeled using the generalized Amber force field, and the protein structures were modeled using the Amber14SB force field [51]. The protein-ligand complexes were solvated with TIP3P water molecules in a cubic box with a length of 12.0 Å, and sodium or chloride ions were added to neutralize the systems. For all the simulations, all the bonds that involved hydrogen atoms were constrained and the Langevin dynamics thermostat was applied to keep the temperature at 300 K.

## Cryo-EM sample preparation, data collection, and structure determination

Purified IadD/E complex was incubated with 20-fold molar excess IAA. Aliquots of 4 μL samples were applied to glow-discharged holey carbon girds (Cu, R1.2/1.3, 300 mesh, Quantifoil). The grids were blotted with force 2 for 3 s and plunged into liquid ethane using a Vitrobot (FEI Thermo Fisher). Cryo-EM data were collected with a Titan Krios microscope (FEI) operated at 300 kV and images were collected using EPU at a nominal magnification of 105,000× (resulting in a calibrated physical pixel size of 0.85 Å/pixel) with a defocus range from −1.2 μM to −2.2 μM. The images were recorded on a K3 summit electron direct detector (Gatan). A dose rate of 15 electrons per pixel per second and an exposure time of 2.5 s were used, generating 40 movie frames with a total dose of approximately 54 electrons per Å2. A total of 2,122 movie stacks were collected (**S2 Table**).

The movie frames were imported to RELION-3 [52]. Movie frames were aligned using MotionCor2 [53] with a binning factor of 2. Contrast transfer function (CTF) parameters were estimated using Gctf [54]. Particles were auto-picked and extracted from the dose-weighted

micrographs. The following steps, including 2D classification, generation of initial model, 3D classification, and 3D refinement, were performed in cryoSPARC [55]. A total of 1,717,755 particles were selected after 2D classification for further processing. Heterogeneous refinement was performed with the 3 initial models to distinguish different conformational states. 3D refinement was performed with C1 symmetry for the selected class representing the intact complex, after confirming that the IAA molecule is present in all the copies of the IadD/E complex. A second round of 3D refinement with C3 symmetry was performed, converging at a resolution of 2.59 Å.

The map was sharpened using Phenix Autosharpen tool [56–58]. The apo IadD/E crystallographic structure (**S1 Table**) was used as the initial template. Manual adjustments and refinements were performed with Coot [45]. Iterative real-space refinement was then performed in Phenix using phenix.real_space_refine [57]. The quality of the model was analyzed with MolProbity in Phenix [59]. Refinement statistics are summarized in **S2 Table**.

## Supporting information

**S1 Fig. Interaction between Iad proteins and IAA (related to Figs 1 and 3).** (**A**) Schematics for the auxin IAA-degradation (*iad*) operon in *Variovorax paradoxus* CL14. The green box highlights the 10 genes examined in the study for IAA transformation. (**B**) ITC measurement of IAA binding to iadk3. "ND" indicates no binding detected. (**C**) Thermal shift assay for iadk3 with IAA and different analogs including 4Cl-IAA, IPA, IBA, 2,4-D, and NAA. Three replicates of each TSA experiments were performed. (**D**) Integrated heat plots for ITC measurements of different Iad proteins binding with IAA. Measurements for proteins iadf-J and iadc with IAA are displayed. ND: no binding detected. Three or more independent measurements were performed. (**E**) Tm changes for Iad proteins in the presence of IAA in the thermal shift assay. Three replications of each TSA experiments were performed. While adding of IAA substantially increased the melting temperature of iadk2, no profound effects were observed for the rest. Source data for **B–E** can be found in **S1 Data**.
(TIF)

**S2 Fig. ITC measurements of PAA to IadK2 and IadK3 (related to Figs 1 and 2).** (**A**) Integrated heat plots for ITC measurement of iadk2 binding with PAA. ND: no binding detected. (**B**) Integrated heat plots for ITC measurements of IadK3 binding with PAA. ND: no binding detected. Two or more measurements were performed. Source data can be found in **S1 Data**.
(TIF)

**S3 Fig. Comparison of IadK2, IadK3, and B-III SBPs (related to Fig 2).** (**A**) The overall structure and the calculated IAA-binding pocket of iadk2 using docking and MD simulation. The L2 linker is colored in pink and the rest is in pale green. Key residues interacting with IAA are shown in sticks representation. (**B**) Structure of iadk3 predicted using alphafold2. The L2 linker is colored in green and the rest is in cyan. The counter residues of iadk2 involved in IAA binding iniadk3 are indicated. Residues labeled in magenta are not conserved between iadk3 and iadk2, which could be related to the IAA specificity of iadk2. (**C**) The closest structural homolog of iadk2 according to the Dali search. The structure of an SBP from *Rhodopseudomonas palustris* in complex with caffeic acid (DHC) is displayed (PDB: 4FB4). The structure containing 2 α helices in L2 belongs to the B-III subcluster of sbps. Key residues involved in substrate DHC binding are shown. (**D**) SAXS analysis of iadk2 complexed with IAA. Left panel: overlaid scattering pattern of the SAXS ab initio model (pink line), the theoretical scattering profiles of iadk2-IAA complex structure from MD simulation (green line) and iadk2 apo crystal structure (blue line) with SAXS scattering data of iadk2-IAA complex. The inserted

panel displays the P(r) distance distributions. Right: Superposition of the ab initio SAXS model (shown as sphere) and the iadk2-IAA complex structure from MD simulation. Source data for **D** can be found in **S1 Data**. (**E**) Multiple sequence alignment. Sequences of iadk2, iadk3, and B-III sbps including Bp0622 (Uniprot Q7VS30), Rpb4630 (Uniprot Q2IR47), Rpd1889 (Uniprot Q139W5), and Rpa1789 (Uniprot Q6N8W4) were aligned with Clustal Omega. Essential residues for iadk2 IAA binding are marked with the filled triangles, among which the strictly conserved residues were shaded in blue, whereas moderately and less conserved residues were in yellow and light green, respectively.
(TIF)

**S4 Fig. Biochemical and structural characterization of IadD/E complex (related to Fig 3).** (**A**) ITC measurements of IAA binding to iadd or iade protein alone. ND: no binding detected. Left panel: no binding between iade and IAA was detected. Right panel: a fitted Kd value of 24 μM was obtained for iadd and IAA. Source data for **A** can be found in **S1 Data**. (**B**) Size-exclusion chromatograms of iadd and iade. The elution fractions from the complex peak were analyzed by SDS-PAGE gel (lower panel). Uncropped SDS-PAGE gel image can be found in **S1 Raw** Image. (**C**) Sedimentation coefficient distributions of iadd/E complex.
(TIF)

**S5 Fig. Structural comparison of apo and IAA-bound IadD/E complex (related to Fig 3).** (**A**) Electron densities and the atomic model around the active site of apo iadd/E. Densities and models around [2Fe-2S], Fe and the substrate-binding pocket were shown. (**B**) Electron densities and the atomic model around the active site of IAA-bound iadd/E. (**C**) Structural comparison of the apo and IAA-bound iadd/E structures. For clarity, 1 heterodimer subunit is displayed. The apo structure is colored in white and IAA-bound structure is colored by proteins. No profound overall conformational changes were identified between the apo and the IAA-bound structures. (**D**) A zoom in view of **c** around the active site. The side chain of H311 in IadD is relocated to accommodate the binding of IAA. Fe and coordinating residues also move slightly inwards (about 1.6 Å) following the binding of IAA.
(TIF)

**S6 Fig. Structural comparison of IadD/E and IadC with homolog structures (related to Figs 3 and 4).** (**A**) Structural comparison of iadd/E-IAA with the homolog structure (PDB: 5AEW). For clarity, a single heterodimer is displayed. Overall, the 2 structures superimposed well with each other, despite some difference in the relative orientation of the α and β subunits. The inset panel in the right shows the detailed insight into the superimposed substrate-binding pocket. (**B**) The homolog structure of iadc (PDR, PDB: 2PIA). The structure of PDR is also composed of 3 major domains, NBD, FBD, and fesd. Ligands FMN and the [2Fe-2S] cluster are indicated. (**C**) Structural comparison of IadC with PDR. PDR is colored in white and IadC is colored by domains. The arrows indicate the relative domain rotations of IadC in comparison to PDR.
(TIF)

**S7 Fig. Cryo-EM data processing for IadD/E in complex with IAA (related to Fig 3).** (**A**) A representative raw cryo-EM micrograph. Scale bar: 50 nm. (**B**) Representative 2D class averages. (**C**) 3D classification. Class 1 represent the bad reconstruction. Class 2 represents the broken protein complex. Class 3 represents the intact protein complex. (**D**) The final cryo-EM map of 3D refinement on Class 3. (**E**) Viewing distribution of the 3D refinement. (**F**) The Fourier shell correlation (FSC) curve of the reconstruction. The 0.143 gold standard FSC cutoff was used to determine the final resolution.
(TIF)

**S8 Fig. Transformation of IAA and analogs by IadC-IadD/E (related to Figs 4 and 5).** (**A**) Color development of IAA and analogs with the Salkowski reagent before and after treatments with iadc-iadd/E. (**B**) Quantification of the relative amounts of IAA or analogs before and after the oxidation reaction catalyzed by the iadc-iadd/E system. Three replications of each condition were performed. Data are presented as mean ± SEM. Source data for **B** can be found in **S1 Data**.
(TIF)

**S9 Fig. IAA metabolite analysis by HPLC (related to Fig 5).** The reacted mixture (RM) was analyzed with HPLC. Analyses for NADH, NAD, IAA, and oxIAA were also performed to assign the peaks.
(TIF)

**S10 Fig. HRMS spectrum of isolated degradation product (oxIAA) from the isotopic experiment (related to Fig 5).** The in vitro IAA degradation reaction with IadC-IadD/E was carried out in the solution containing 95% $H_2O^{18}$ (v/v).
(TIF)

**S11 Fig. IAA transformation by *E. coli* transformed with *IadK2/C/D/E* genes (related to Fig 6).** Effects of Iad proteins mutants in the in vivo IAA transformation by *E. coli*. Two vectors, *v1* (containing genes *IadK2* and *IadC*) and *v2* (with genes *IadD* and *IadE*), were transformed to *E. coli* BL21(DE3) to enable IAA transformation. Mutants of IadK2 (W172L, R199A), IadD (H221A), and IadC (S61A) were tested. Little IAA conversion was observed with the empty vector without the *iad* genes or with the loss-of-function mutations in *iad*C or *iad*D. Source data can be found in **S1 Data**.
(TIF)

**S1 Table. Crystallographic data collection and refinement statistics.**
(DOCX)

**S2 Table. Cryo-EM data collection, refinement, and validation statistics.**
(DOCX)

**S1 Data. Source data for graphs in this paper.**
(XLSX)

**S1 Raw Image. Uncropped SDS-PAGE gel image in this paper.**
(PDF)

## Acknowledgments

We thank the staffs from the BL17B/BL18U1/BL19U1/BL19U2/BL01B beamlines of the National Facility for Protein Science (NFPS) at Shanghai Synchrotron Radiation Facility for assistance in crystallographic data collection. The cryo-EM data was collected using the Cryo-Electron Microscopy Facility of Hubei University. We thank Prof. Jeffery L. Dangl for critical reading and useful comments.

## Author Contributions

**Conceptualization:** Guimei Yu, Heng Zhang.

**Formal analysis:** Guimei Yu, Heng Zhang.

**Funding acquisition:** Zhuang Li, Huabing Sun, Guimei Yu, Heng Zhang.

**Investigation:** Yongjian Ma, Xuzichao Li, Feng Wang, Lingling Zhang, Shengmin Zhou, Xing Che, Dehao Yu, Xiang Liu, Zhuang Li, Huabing Sun, Heng Zhang.

**Methodology:** Huabing Sun, Guimei Yu, Heng Zhang.

**Project administration:** Heng Zhang.

**Resources:** Huabing Sun.

**Software:** Huabing Sun.

**Supervision:** Guimei Yu, Heng Zhang.

**Writing – original draft:** Guimei Yu.

**Writing – review & editing:** Guimei Yu, Heng Zhang.

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
