## [Editor Report · Decision Letter 0]

19 Dec 2022

Dear Dr Yu, 

Thank you for submitting your manuscript entitled "Molecular Mechanism for Bacterial Degradation of Plant Hormone Auxin" for consideration as a Research Article by PLOS Biology.

Your manuscript has now been evaluated by the PLOS Biology editorial staff, as well as by an academic editor with relevant expertise, and I am writing to let you know that we would like to send your submission out for external peer review.

Once your full submission is complete, your paper will undergo a series of checks in preparation for peer review. After your manuscript has passed the checks it will be sent out for review. To provide the metadata for your submission, please Login to Editorial Manager (https://www.editorialmanager.com/pbiology) within two working days, i.e. by Dec 21 2022 11:59PM.

Kind regards,

Richard

Richard Hodge, PhD

Associate Editor, PLOS Biology

rhodge@plos.org

PLOS

---

## [Decision Letter · Decision Letter 1]

9 Feb 2023

Dear Dr Yu,

Thank you for your patience while your manuscript "Molecular Mechanism for Bacterial Degradation of Plant Hormone Auxin" was peer-reviewed at PLOS Biology. Please accept my apologies for the delays that you have experienced during the peer review process. Your manuscript has now been evaluated by the PLOS Biology editors, an Academic Editor with relevant expertise, and by three independent reviewers. 

In light of the reviews, which you will find at the end of this email, we would like to invite you to revise the work to thoroughly address the reviewers' reports.

As you will see, the reviewers think the study is interesting and well done, but raise concerns about the lack of comparison with a previous study in Nature Microbiology that recently characterized an iac/iad auxin degradation locus (Conway et al, 2022, PMID 36266335), as well as noting that different nomenclature appears to be used to define the operon (adg vs iac). I have provided some specific comments from the Academic Editor on this point below the reviewer reports, and we ask that you please clarify whether your study is focusing on the same operon where a different reaction and degradation product are described, or whether the paper focuses on a distinct operon. In addition, the reviewers also ask for several additional control experiments, including a mutational validation in AdgA to provide further support for the IAA binding model and using PAA in the ITC assays to strengthen the conclusions regarding substrate specificity.

Given the extent of revision needed, we cannot make a decision about publication until we have seen the revised manuscript and your response to the reviewers' comments. Your revised manuscript is likely to be sent for further evaluation by all or a subset of the reviewers.

**IMPORTANT - SUBMITTING YOUR REVISION**

*Re-submission Checklist*

*Published Peer Review*

*PLOS Data Policy*

*Blot and Gel Data Policy*

Sincerely,

Richard

Richard Hodge, PhD

Associate Editor, PLOS Biology

rhodge@plos.org

REVIEWS:

Reviewer #1: The manuscript by Ma, et al. describes the ligand binding, structure and activity of proteins from the Variovorax auxin-degradation operon responsible for IAA binding and degradation. The authors performed crystallization studies and enzymatic assays to demonstrate IAA binding and enzymatic activity by AdgB, I, J and H. The degradation product of IAA catalysed by AdI/H/J was further characterized by mass spectrometry. The genes responsible for IAA degradation were then tested in E. coli and IAA degradation tested. Understanding the enzymes involved in IAA degradation and the crosstalk between plants and bacteria is of broad interest to the plant biology community.

Certain points need to be addressed to better clarify the results of the manuscript and to put it into a broader context of what is known about IAA degrading bacteria-

The results need to be discussed in context of recent literature, in particular Conway, et al. 2022 https://doi.org/10.1038/s41564-022-01244-3 The catalytic mechanisms and different reaction products should be explained. As the nomenclature is different, it is difficult for the reader to follow. Are the authors using V. paradoxus in this study? Is this the same operon as described in Conway, et al.?

I am unclear as to the applications the authors envision for this operon in engineering the rhizosphere microbiome. This would be making transgenic soil bacteria to alter RGI? 

ITC measurements need s.d. or s.e.m. How many times were these measurements performed? For thermal shift assays in Fig. 1 the error bars are very faint and hard to see.

Given that AdgB is not in complex with IAA, docking studies were done. SAXS could be used to determine if AdgB adopts closed conformation upon IAA binding as co-crystallisation and soaking did not work.

Mutations of AdgA to bind IAA based on the structural and modelling studies of AdgB would be interesting to further support the model for IAA specific binding. 

The structures in Fig.3 examine both X-ray and EM structures with the X-ray structure of AdgH/I in the apo form and the EM structure in the IAA bound form. A larger portion of the EM map should be shown around the active site and more detailed comparison of the conformation of the side chains in the apo and ligand bound form described. How well-positioned are the side chains in the EM model vs. the X-ray structure?

Fig. 4 -The schematic illustration is too small and uninterpretable. It is unclear where the electron is coming from and where it is going and how the reaction is catalysed.

For the crystal tables, please provide CC1/2, Ramachandran statistics should be included. The clash scores and R factors are quite high for 7YLT. Is this structure fully refined? Also there seem to be a lot of RSTZ outliers for all structures.

Supp Fig. 3 The SAXS experimental curve and theoretical model are very different. The fit looks extremely poor, especially at high q range, can the authors address this? 

Reviewer #2: This manuscript is very straightforward, and summarizes an impressive amount of work, including the biochemical characterization of three different enzymes involved in IAA transport and metabolism. Overall, the claims are well supported. The findings are important, as they increase our understanding of how microbes can alter the concentration of the plant hormone IAA, an auxin, in plant roots and the rhizosphere.

The manuscript is succinctly written, and in some places additional key information and details need to be provided so assist the readers in understanding the experiments and to strengthen the rationale for the work. Additionally, I believe use of the term "degradation" is often inappropriately used, and thus confusing to the reader. 

Scientific comments and concerns 

1. An IAA degradation locus referred to as iad has been recently described (Conway et al, 2022 Nature Microbiology). Is the adg locus discussed here distinct from that locus? 

2. The naturally occurring auxin phenyl acetic acid (PAA) should be included in the ITC experiments investigating substrate specificity. This is important to examine before the authors can claim that the protein has specificity for bicyclic compounds.

3. I was confused by the use of the term degradation, when the authors actually mean transformation (or conversion) of IAA into oxIAA, which is presumably inactive, but still present in the cell. 

4. It seems the authors believe that the SBP proteins are required for entry of IAA into the cell. However, at acidic pH (e.g. pH 5.8) much of the IAA is protonated, and thus may be able to diffuse across the membrane without the aid of transporters (at least this is what plant biologists who study IAA assume). It would be very interesting to test this idea in a Varivorax adgB mutant. This question may be beyond the scope of this manuscript, but could be addressed in future studies.

5. The question about pH and entry of IAA into bacteria cells arose when I was considering the experiment shown in Fig. 6 (ecotpic expression of adg genes in E. coli). Presumably these experiments were done in pH7 media, but it would be helpful to confirm that the LB used in the IAA degradation assays was at pH 7 .

6. A control is missing for the experiment shown in Fig. 4 B. How can we ensure that the colormetric change does not occur spontaneously given enough time? A "no enzyme added" negative control is needed. Also, what is the time frame shown for the panel on the right, compared to the graph on the left? 

7. More information regarding what is shown on the y axes in fig. 4 would be helpful. 

Editorial comments

Introduction:

1. I believe the PAA, a natural and active form of auxin, is more abundant than IAA in many plants. 

2. Is reference 9 the appropriate citation for the general statement that "most rhizobacteria are found capable of producing IAA"? Referencing a review article would be more suitable. 

8. An IAA degradation locus referred to as iad has been recently described (Conway et al, 2022 Nature Microbiology). Are the adg genes here distinct from that locus? 

9. Is it appropriate to refer to conversion of IAA to oxIAA as degradation? I believe the term "inactivation" might be more appropriately used throughout the manuscript, until the discussion section, when a larger picture is provided.

10. Figure 5 legend: 

a. panel A- it would be helpful to indicate the color of the traces for before and after the transformation

b. Panel c: what concentration of compounds are used in the experiment shown? 

Results

11. Page 4, line the term "predicted" should be used when referring to products of genes that have not been characterized yet. The adg operon IS PREDICTED to encode….

Methods:

12. The Genbank accession numbers should be provided so that readers can analyze the genes/proteins

Reviewer #3: This study builds on an earlier report by Finkel et al., 2020, which defined an auxin (IAA) degradation locus in the bacterium Variovorax paradoxus. Here the authors define (1) the structure and auxin binding mechanism for the ABC transporter solute binding protein AdgB and (2) a structure of an AdgH/AdgI non-heme Rieske dioxygenase complex. The authors further demonstrate that AdgH/I/J can catalyze IAA degradation to oxIAA. Overall, this is a well designed/executed study that combines quantitative biochemistry, structural biology and enzymology to characterize a novel bacterial IAA degradation pathway. The crystal and cryoEM structure show good sterochemistry and refinement statistics. As a non-expert in this field, I had however some difficulties to compare this work to earlier studies. In addition, there are some minor technical issues worth considering. I will summarize my points below:

1. MAJOR: A recent study by Conway et al, 2022 (https://doi.org/10.1038/s41564-022-01244-3) reported the gene products iadCDE as the minimal functional unit required for IAA degradation in Variovorax, and the regulation of root growth inhibition of a plant model (compare Fig. 1c in this study). Do the authors use the same nomenclature in this work (Fig. 1A) and if so, does that mean that there are two auxin degradation pathways encoded in Variovorax gene cluster, one represented by iad/adgCDE and one by adgH/I/J? If so, why are iadD and iadE necessary and sufficient for IAA degradation and root growth inhibition mediated by Variovorax (Conway et al, 2022, Fig. 1d)?

2. MAJOR: I understand that iadB/adgB is the bacterial ABC transporter enabling IAA uptake into the cytoplasm of Variovorax. Assuming that all other degradation enzymes are present in the cytoplasm, does that mean that adgB is essential for both the iadC/D/E and the adgH/I/J pathways? Has this been experimentally tested, for example by genetic epistasis analysis?

3. MINOR: As anticipated, AdgB could engage IAA with nanomolar affinity, whereas no obvious binding was detected for AdgA and IAA. Why was it anticipated that AdgA would not bind IAA? This could be discussed later in the text, when the authors compare the degree of conservation of the putative IAA binding pocket in both proteins.

4. MAJOR: Fig. 1C, can the authors rationalize the bimodal binding behavior for recombinant AdgB in the ITC assay? The structure seems to indicate a single IAA binding site?

5. MAJOR: Conway et al, 2022 et al reported iad/C/D/E to form an IAA degradation complex. Have these proteins been tested in the ITC assays shown in Supplementary Fig. 1, and if so, why was no binding detected? It seems that MarR domains, that show high affinity IAA binding (Kd ~0.3 uM) (Conway et al. 2022) were excluded from the analysis in Fig. S1? This should be mentioned in the figure legend.

6. MAJOR: In the Figure 5, the authors nicely demonstrate that AdgH/I/J can catalyze IAA oxidation. Can these data be integrated with the metabolic analysis by Conway et al, 2022, Fig. 4? Could this imply that iadCDE and adgHIJ are part of one degradation pathway, and that adgHIJ may use 2-hydroxy-indole-3-acetic acid or dioxindole-3-acetic acid (DOAA) as substrates in vivo? Please clarify.

7. MINOR: Figure 5a/Figure S7. Could the HPLC buffer system or column be optimized to obtain a sharper peak for IAA?

8. MAJOR: Fig. 5. I would recommend to determine rate constants for the different substrates.

9. MINOR: Fig. 6D may need to be revised depending on the outcome of the analysis on point 6 (see above)

10. MINOR: Introduction: "Elucidating the underlying mechanism of bacterial IAA degradation would help to maximize the beneficial effects of auxin degradation for ecological agriculture." How would that be achieved? 

11. MINOR: Introduction: "Dissecting and defining the functional mechanism of the adg operon is therefore of significant importance for the applications in ecological and stress agricultures." What type of application exist that make use of bacterial IAA degradation, please specify.

*COMMENTS FROM THE ACADEMIC EDITOR*

I agree with the reviewers that the lack of comparison the Conway is a major weakness and it isn’t clear where the “adg” nomenclature/operon (instead “iac” in the Finkle 2020 paper and iac/iad in the Conway 2022 paper) came from. Similarly, the operon structures in Conway seems a bit different than in this paper so I’m wondering if these are distinct operons. However, if that’s the case, then the authors of this paper should clearly state how they decided to study the adg operon and it’s demonstrated role in auxin degradation (and if it is different from or the same as iac or iad).

Futhermore, the IAA degradation reactions described are different. The Conway paper describes that the iac operon breaks down IAA through DOAA to catechol while the iad locus converts IAA to anthranilic acid. In this paper, they describe oxIAA as the next step in degradation by the adg degrading operon. Either Ma et al. are describing the same operon but found a different reaction and degradation product or they’re describing a distinct operon suggesting 3 Variovorax degrading pathways. Importantly, I couldn’t find the species or strain listed anywhere in the materials and methods so it’s not clear if this is the same strain. This definitely needs to be clarified and either outcome (two mechanisms, or inconsistency with previous papers) should be clarified.

---

## [Editor Report · Decision Letter 2]

1 Jun 2023

Dear Dr Yu,

Thank you for your patience while we considered your revised manuscript "Molecular Mechanism for Bacterial Degradation of Plant Hormone Auxin" for publication as a Research Article at PLOS Biology. This revised version of your manuscript has been evaluated by the PLOS Biology editors and the Academic Editor.

Based on our Academic Editor's assessment of your revision, I am pleased to say that we are likely to accept this manuscript for publication, provided you satisfactorily address the following data and other policy-related requests that I have provided below (A-F):

(A) We would like to suggest the following modification to the title: 

“Structural and biochemical characterization of the key components of an auxin-degradation operon from the rhizosphere bacterium Variovorax”

(B) You may be aware of the PLOS Data Policy, which requires that all data be made available without restriction: http://journals.plos.org/plosbiology/s/data-availability. For more information, please also see this editorial: http://dx.doi.org/10.1371/journal.pbio.1001797

- Supplementary files (e.g., excel). Please ensure that all data files are uploaded as 'Supporting Information' and are invariably referred to (in the manuscript, figure legends, and the Description field when uploading your files) using the following format verbatim: S1 Data, S2 Data, etc. Multiple panels of a single or even several figures can be included as multiple sheets in one excel file that is saved using exactly the following convention: S1_Data.xlsx (using an underscore).

- Deposition in a publicly available repository. Please also provide the accession code or a reviewer link so that we may view your data before publication. 

Figure 1C-E, 2D-E, 3A, 3F, 4A-B, 4D, 4F, 5A-B, 6B-C, S1B-E, S2A-B, S3D, S4A, S8B, S11

(C) Thank you for depositing the structural data in the PDB (7YLT, 7YLS, 7YLR, 8H2T) and EMDB databases (EMD-34443). However, I note that the data is currently on hold. We ask that you please make this data publicly available before publication.

(D) Please also ensure that each of the relevant figure legends in your manuscript include information on *WHERE THE UNDERLYING DATA CAN BE FOUND*, and ensure your supplemental data file/s has a legend.

(E) We require the original, uncropped and minimally adjusted images supporting all blot and gel results reported in the following Figures:

Figure S4B

We will require these files before a manuscript can be accepted so please prepare and upload them now. Please carefully read our guidelines for how to prepare and upload this data: https://journals.plos.org/plosbiology/s/figures#loc-blot-and-gel-reporting-requirements

(F) Please ensure that your Data Statement in the submission system accurately describes where your data can be found and is in final format, as it will be published as written there. This includes referencing where the underlying data can be found in the Supplementary Information. 

We expect to receive your revised manuscript within two weeks. 

*Published Peer Review History*

*Press*

Kind regards,

Richard

Richard Hodge, PhD

Associate Editor, PLOS Biology

rhodge@plos.org

PLOS

---

## [Editor Report · Decision Letter 3]

8 Jun 2023

Dear Dr Yu,

On behalf of my colleagues and the Academic Editor, Cara Haney, I am pleased to say that we can accept your manuscript for publication, provided you address any remaining formatting and reporting issues. These will be detailed in an email you should receive within 2-3 business days from our colleagues in the journal operations team; no action is required from you until then. Please note that we will not be able to formally accept your manuscript and schedule it for publication until you have completed any requested changes.

PRESS

Kind regards, 

Richard

Richard Hodge, PhD

rhodge@plos.org

PLOS
